# Under-reported relationship: a comparative study of pharmaceutical industry and patient organisation payment disclosures in the UK (2012–2016)

Piotr Ozieranski [iD] ,[1] Marcell Csanádi,[2] Emily Rickard,[1] Shai Mulinari[3]

¹Social and Policy Sciences, University of Bath, Bath, UK
²Syreon Research Institute, Budapest, Hungary
³Sociology, Lunds Universitet, Lund, Sweden

**Correspondence to**
Dr Piotr Ozieranski;
p.ozieranski@bath.ac.uk

## ABSTRACT

**Objectives** To examine the under-reporting of pharmaceutical company payments to patient organisations by donors and recipients.
**Design** Comparative descriptive analysis of payments disclosed on drug company and charity regulator websites.
**Setting** UK.
**Participants** 87 donors (drug companies) and 425 recipients (patient organisations) reporting payments in 2012–2016.
**Main outcome measures** Number and value of payments reported by donors and recipients; differences in reported payments from/to the same donors and recipients; payments reported in either dataset but not the other one; agreement between donor–recipient ties established by payments; overlap between donor and recipient lists and, respectively, industry and patient organisation data.
**Results** Of 87 donors, 63 (72.4%) reported payments but 84 (96.6%) were mentioned by recipients. Although donors listed 425 recipients, only 200 (47.1%) reported payments. The number and value of payments reported by donors were 259.8% and 163.7% greater than those reported by recipients, respectively. The number of donors with matching payment numbers and values in both datasets were 3.4% and 0.0%, respectively; for recipients these figures were 7.8% and 1.9%. There were 24 and 3 donors missing from industry and patient organisation data during the entire study period, representing 38.1% and 3.6% of those in the respective datasets. The share of donor–recipient ties in which industry and patient organisation data agreed about donors and recipients was 38.9% and 68.4% in each dataset, respectively. Of 63 donors reporting payments, only 3 (4.8%) had their recipient lists fully overlapping with patient organisation data. Of 200 recipients reporting industry funding, 102 (51.0%) had their donor lists fully overlapping with industry data.
**Conclusions** Both donors and recipients under-reported payments. Existing donor and recipient disclosure systems cannot manage potential conflicts of interest associated with industry payments. Increased standardisation could limit the under-reporting by each side but only an integrated donor–recipient database could eliminate it.

### Strengths and limitations of this study

► We examine the under-reporting of pharmaceutical industry payments to patient organisations using large samples of donors and recipients over a period of time.
► We systematically compare the under-reporting by donors and recipients using five complementary measures at different levels of analysis.
► One key limitation is that the samples of donors and recipients were not exhaustive.
► The full extent of under-reporting remains unknown as no definitive list of payments exists.

## INTRODUCTION

Many patient organisations accept funding from drug companies. A recent UK analysis showed that the industry donated over £57 million (€65 million; $73 million) to 508 patient organisations, with the annual sum more than doubling from 2012 to 2016.[1] Another study of 289 US patient organisations found that 156 (67%) received funding from for-profit companies, with a median proportion of 45% of their income coming from drug, medical device or biotechnology companies.[2] Although industry funding may benefit patient organisations,[3 4] it raises concerns about potential conflict of interests (COIs) compromising patient organisations' independence and credibility.[5–10]

Both donors[11] and recipients[4] assert that any COIs can be managed by careful disclosure of funding. On the donor side, since 2012 members of pharmaceutical industry trade groups in European countries,[12] such as the Association of the British Pharmaceutical Industry (ABPI), have disclosed payments to patient organisations annually on each company's website, including their monetary value and purpose.[13] In the UK,

the number of companies subscribing to the ABPI Code also includes over 60 non-members of the ABPI; hence, nearly every relevant company is covered.[14] However, the self-regulatory approach to payment disclosure has short-comings, including absent reports, unclear or inadequate payment descriptions and unstandardised reporting.[1 15] Likewise, on the recipient side, disclosures published on patient organisations' websites have been criticised as incomplete or uninformative.[16–20] In the UK, a possibly more reliable, yet rarely examined,[15 21] source of disclosures are mandatory annual accounts of patient organisations registered as charities with an annual income over £25 000. These accounts serve a dual-purpose of detailing the charity's activities and ensuring financial transparency.

Notwithstanding the shortcomings of each side's disclosures, and the potentially complementary information they provide, donor and recipient disclosures are rarely compared. One exception is a recent UK study which identified companies not disclosing payments to patient organisations being mentioned in their annual accounts as donors.[15] Another UK study found discrepancies between annual accounts of some patient organisations contributing to health technology assessment in England and drug company payment disclosures.[21] Similarly, discrepancies were found between sponsorships reported on patient organisation and drug company websites in Italy.[19] More broadly, cross-interrogation of different data sources has revealed undisclosed industry ties among treatment guideline panellists,[22–24] clinical trialists, authors of medical journal articles[25–29] and some clinical commissioning groups[30] and National Health Service trusts in England.[31]

We examine the under-reporting of payments to UK patient organisations from 2012 to 2016 by comparing payment disclosures made by 87 companies to 425 patient organisations with the annual accounts of the same set of patient organisations. Specifically, by considering the extent to which the industry and patient organisation disclosures differ regarding who provided and received funding, and how much was paid and received, we examine the reliability of the two disclosures systems said to neutralise concerns about COIs.

## METHODS
### Data sources and extraction
Absent of a complete list of non-ABPI member companies that subscribe to the ABPI ode, our sample of drug companies comprised all 108 participants of Disclosure UK in 2015, including 53 ABPI members and 55 non-members (online supplementary file 1). Disclosure UK is a self-regulatory initiative run by the ABPI, covering payments to healthcare organisations[32] and professionals,[33] but because its participants subscribe to the ABPI Code, they should also separately disclose, on their websites, payments to patient organisations once a year.[1 15]

In June 2017 and January 2018, ER searched the websites of the 108 companies, identifying 220 disclosure reports published between 2012 and 2016. ER downloaded them and extracted payment data into a single Excel spreadsheet, converting all payment values to 2016 sterling using inflation data from the Office for National Statistics.[34] There were 66 (61.1%) companies disclosing payments in at least 1 year during this period. Following the exclusion of ineligible payments based on their timing or recipient characteristics, we established that 64 companies reported 4572 payments, worth £57 305 289.2 to 489 UK patient organisations (online supplementary file 1).

In September 2017 and July 2018, ER searched for these patient organisations on the websites of the UK charity regulators—the Charity Commission for England and Wales, the Scottish Charity Regulator and the Northern Irish Charity Regulator—to which registered charities must submit their annual accounts. We considered accounts from financial years ending in 2012–2017 to cover all calendar year data from drug company reports. We found that 425 (86.9%) patient organisations drug companies listed as recipients published their annual accounts at least once with at least one charity regulator, with a total of 1428 annual accounts. In industry data, 63 (98.4%) companies reported 4316 (94.4%) payments, worth £54 071 454.21 (94.4%) to these patient organisations.

ER read all annual accounts and extracted any information pertaining to drug company payments to an Excel spreadsheet. As with industry data, if several values were mentioned in a single payment description they were considered as separate payments. As patient organisation yearly accounts a lack specific monthly end date,[21] we turned those covering months up to June into the previous calendar year, and the rest into the current calendar year. The 425 patient organisations registered as charities reported 4372 payments, worth £58 668 293.6, from 167 donors. Following exclusion criteria referring to donor and recipient characteristics as well as the conversion of financial into calendar years (online supplementary file 2), we established that 200 (47.1%) organisations reported 1661 (38.0%) payments, worth £33 037 955.8 (56.3%), from 84 (47.1%) drug companies. Notably, we excluded 2014 payments, worth £1 992 147.3, made by 61 companies found in patient organisation data but missing from the original sample of Disclosure UK participants.

One researcher, ER, collected drug company disclosure reports and patient organisation annual accounts. Drug company and charity regulator websites are well-structured and ER searched the websites at two points in time, with the second phase of data collection being used to check the accuracy of the initial one. Therefore, we are confident that no relevant documents were missed. When extracting payment data from drug company and patient organisation documents into Excel spreadsheets, ER checked twice that each payment was transferred correctly. As these documents follow a similar format, we deemed it sufficient for another researcher, PO, to check

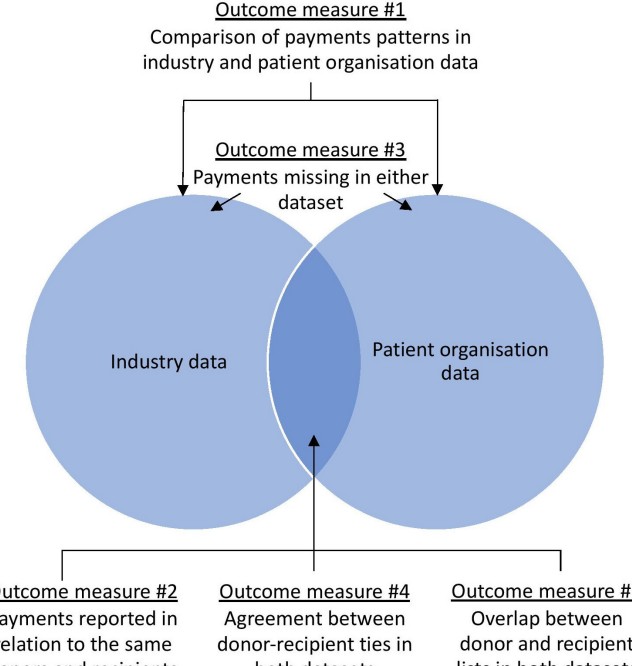

Outcome measure #1
Comparison of payments patterns in industry and patient organisation data

Outcome measure #3
Payments missing in either dataset

Industry data

Patient organisation data

Outcome measure #2
Payments reported in relation to the same donors and recipients

Outcome measure #4
Agreement between donor-recipient ties in both datasets

Outcome measure #5
Overlap between donor and recipient lists in both datasets

**Figure 1** Relationships between the outcome measures of under-reporting of drug company payments in industry and patient organisation data. The Venn diagram illustrates the relationships between the measures of payment under-reporting employed in this paper by showing the overlap between the industry and patient organisation datasets. The size of specific parts of the diagram does not reflect the size of either of the datasets or the overlap between them. All outcome measures were reported for the number and value of payments.

20% randomly selected drug company payment disclosure reports and patient organisation annual accounts with no discrepancies found.

### Analysis
MC and PO converted the data into donor–recipient matrices and analysed it descriptively in Excel. We examined the overall pattern of the number and value of payments in each dataset and its changes over time.

### Outcome measures
Absent an exhaustive list of payments, enabling a direct comparison of industry and patient organisation data, we studied under-reporting with five complementary measures. Our calculations can be verified by consulting online supplementary files 3-20 signposted throughout the Results section. The web supplements are accessible via the University of Bath Research Data Archive in the Excel format.[35]

First, we compared the overall volume of payments reported in industry and patient organisation data, including payments lacking specific values.

Second, we examined the absolute and relative differences in payments reported in relation to the same donors and recipients in industry and patient organisation data. We calculated the absolute difference as the difference

between the number/value of payments reported in the two datasets; the relative difference was the absolute difference divided by the number/value of payments in the dataset with the higher number/value of payments. To compare larger and smaller donors and recipients we considered absolute and relative differences at different thresholds of the overall number (ie, ≥1, >1, >10, >100 in at least one dataset) and value of payments (ie, >£0, >£10 000, >£100 000 in at least one dataset).

Third, focusing on the highest relative differences, in which donors or recipients were present in one but absent from the other dataset, we calculated the number and value of payments related to donors and recipients missing from each dataset.

Fourth, we compared the extent to which industry and patient organisation data conveyed the same pattern of connections between donors and recipients. Specifically, we analysed the overlap of ties (or links) between donors and recipients—established by the presence of payments—in each dataset.

Finally, building on the analysis of donor-industry ties at the level of the two datasets, we moved to the donor and recipient level by examining the extent of overlap between recipient and donor lists in patient organisation and industry data, respectively.

### Patient and public involvement
Neither patient groups nor the public were involved in this study. We plan to disseminate key findings in an accessible format using a blog post.

### RESULTS
The results follow the measures of under-reporting of payments outlined in the Methods section. We summarise relationships between these measures in figure 1.

### Comparison of payment patterns in industry and patient organisation data
Although 63 donors reported payments, 84 were mentioned by recipients. In industry data, donors disclosed payments to 425 recipients but only 200 (47.1%) of those reported payments during the study period (table 1).

The number of payments disclosed by donors was 259.8% greater than those disclosed by recipients (table 1). Each side was disclosing more payments over time, but the increase from 2012 to 2016 was greater in industry data (141.2% vs 115.8%, respectively). In industry data, 4235 (98.1%) payments reported by 62 (98.4%) donors to 416 (97.9%) recipients had values greater than 0. In patient organisation data, only 772 (46.5%) payments reported by 121 (60.5%) recipients from 62 (73.8%) donors had values greater than 0. The value of payments disclosed by donors was 163.7% higher than those disclosed by recipients. Over the 5-year period the value of payments reported by donors increased by 257.6%, while those by recipients—by 434.7%.

**Table 1** Patterns in payments reported by donors and recipients*

| | All years | 2012 | 2013 | 2014 | 2015 | 2016 |
|---|---|---|---|---|---|---|
| **Calculations based on the number of payments** | | | | | | |
| Number of payments—industry data | 4316 | 769 | 706 | 821 | 934 | 1086 |
| Number of payments—patient organisation data | 1661 | 291 | 326 | 346 | 361 | 337 |
| Number of drug companies with at least one payment—industry data | 63 | 30 | 38 | 45 | 51 | 44 |
| Number of drug companies with at least one payment—patient organisation data | 84 | 56 | 65 | 65 | 69 | 65 |
| Number of patient organisations with at least one payment—industry data | 425 | 197 | 198 | 225 | 244 | 244 |
| Number of patient organisations with at least one payment—patient organisation data | 200 | 103 | 102 | 116 | 126 | 117 |
| **Calculations based on the value of payments** | | | | | | |
| Number of payments with value >£0—industry data | 4235 | 748 | 682 | 813 | 915 | 1077 |
| Number of payments with value >£0—patient organisation data | 772 | 144 | 148 | 144 | 173 | 163 |
| Value of payments—industry data (2016 £) | 54 071 454.2 | 7 797 600.4 | 6 189 152.7 | 8 583 101.6 | 11 415 293.2 | 20 086 306.3 |
| Value of payments—patient organisation data (2016 £) | 33 037 955.8 | 3 459 605.6 | 4 414 328.4 | 5 301 728.3 | 4 825 091.2 | 15 037,202.3 |
| Number of drug companies with at least one payment with the value >£0—industry data | 62 | 30 | 38 | 45 | 50 | 44 |
| Number of drug companies with at least one payment with the value >£0—patient organisation data | 62 | 42 | 44 | 42 | 46 | 44 |
| Number of patient organisations with at least one payment with the value >£0—industry data | 416 | 197 | 193 | 225 | 237 | 244 |
| Number of patient organisations with at least one payment with the value >£0—patient organisation data | 121 | 57 | 51 | 58 | 72 | 57 |

*We used the following values of the Consumer Price Index obtained from Office for National Statistics to adjust payment values for inflation: 2012=96, 2013=98.2, 2014=99.6, 2015=100, 2016=101.

**Table 2** Donors—absolute and relative differences in the number and value of payments

| | All years | % |
|---|---|---|
| **Calculations based on the number of payments—absolute differences** | | |
| Number of drug companies with at least one payment in at least one dataset—industry and patient organisation data | 87 | |
| Number of drug companies with exact match in both datasets | 3 | 3.4 |
| Number of drug companies with more payments in industry data | 49 | 56.3 |
| Number of drug companies with more payments in patient organisation data | 35 | 40.2 |
| Highest absolute difference between patient organisation and industry data—number of payments higher in industry data | 757 | |
| Highest absolute difference between patient organisation and industry data—number of payments higher in patient organisation data | 53 | |
| **Calculations based on the number of payments—relative differences** | | |
| Number of drug companies with relative difference <10% | 5 | 5.7 |
| Number of drug companies with relative difference <20% | 9 | 10.3 |
| Number of drug companies with relative difference <50% | 21 | 24.1 |
| Number of drug companies with relative difference=100% | 27 | 31.0 |
| **Calculations based on the value of payments—absolute differences** | | |
| Number of drug companies with at least one payment with value >£0 in at least one dataset | 74 | 100 |
| Number of drug companies with exact match in both datasets | 0 | 0.0 |
| Number of drug companies with higher payment value in industry data | 48 | 64.9 |
| Number of drug companies with higher payment value in patient organisation data | 26 | 35.1 |
| Highest absolute difference between patient organisation and industry data—number of payments higher in industry data (2016 £) | 6 406 351.2 | |
| Highest absolute difference between patient organisation and industry data—number of payments higher in patient organisation data (2016 £) | 2 960 716.0 | |
| **Calculations based on the value of payments—relative differences** | | |
| Number of drug companies with relative difference <10% | 3 | 4.1 |
| Number of drug companies with relative difference <20% | 9 | 12.2 |
| Number of drug companies with relative difference <50% | 14 | 18.9 |
| Number of drug companies with relative difference=100% | 24 | 32.4 |

## Payments reported in relation to the same donors and recipients in industry and patient organisation data

Having considered the overall pattern of payments, we now compare those reported in relation to the same donors and recipients in each dataset.

### Donors

Of the 87 donors identified in industry and patient organisation data, 3 (3.4%) had matching payment numbers, and 49 (56.3%) had more payments in patient organisation data (table 2; online supplementary file 3). In relative terms, only 21 (24.1%) donors had a difference in the number of payments less than 50%, and 27 (31.0%) had a difference of 100%, equivalent to only either donors or recipients reporting payments.

In both datasets, 74 (85.1%) donors had payment values greater than 0 (online supplementary file 3). Of those, 48 (64.9%) had a higher payment value in industry data and the remaining ones—in patient organisation data.

Only 14 (18.9%) donors with payment values greater than 0 had a relative difference lower than 50%, while 24 (32.4%) donors had a relative difference of 100%.

Overall, donors with a high overall number and value of payments in either dataset typically had high absolute differences (online supplementary file 3). Contrastingly, donors with a small number or value of payments usually had higher relative differences, that is, high relative differences often resulted from few payments or payments of small value in one dataset but none in the other.

Over time, there was little improvement in the shares of donors with matching numbers of payments (from 1.6% in 2012 to 5.5% in 2016); the share of donors with matching payment values showed no improvement (0%). However, the shares of those with relative differences in payment numbers or values lower than 50% increased (from 11.5% to 17.4%; and from 7.7% to 25.5%, respectively) (online supplementary file 4).

**Table 3** Recipients—absolute and relative differences in the number and value of payments

| | All years | % |
|---|---|---|
| **Calculations based on the number of payments—absolute differences** | | |
| Number of patient organisations with at least one payment in at least one dataset—industry and patient organisation data | 425 | |
| Number of patient organisations with exact match in both datasets | 33 | 7.8 |
| Number of patient organisations with more payments in industry data | 335 | 78.8 |
| Number of patient organisations with more payments in patient organisation data | 57 | 13.4 |
| Highest absolute difference between patient organisation and industry data—number of payments higher in industry data | 199 | |
| Highest absolute difference between patient organisation and industry data—number of payments higher in patient organisation data | 33 | |
| **Calculations based on the number of payments—relative differences** | | |
| Number of patient organisations with relative difference <10% | 35 | 8.2 |
| Number of patient organisations with relative difference <20% | 42 | 9.9 |
| Number of patient organisations with relative difference <50% | 81 | 19.1 |
| Number of patient organisations with relative difference=100% | 225 | 52.9 |
| **Calculations based on the value of payments—absolute differences** | | |
| Number of patient organisations with at least one payment with value >£0 in at least one dataset | 416 | 100 |
| Number of patient organisations with exact match in both datasets | 8 | 1.9 |
| Number of patient organisations with higher payment value in industry data | 356 | 85.6 |
| Number of patient organisations with higher payment value in patient organisation data | 52 | 12.5 |
| Highest absolute difference between patient organisation and industry data—number of payments higher in industry data (2016 £) | 6 718 576.6 | |
| Highest absolute difference between patient organisation and industry data—number of payments higher in patient organisation data (2016 £) | 6 493 237.1 | |
| **Calculations based on the value of payments—relative differences** | | |
| Number of patient organisations with relative difference <10% | 21 | 5.0 |
| Number of patient organisations with relative difference <20% | 34 | 8.2 |
| Number of patient organisations with relative difference <50% | 73 | 17.5 |
| Number of patient organisations with relative difference=100% | 295 | 70.9 |

## Recipients

Of the 425 recipients identified in industry data, 33 (7.8%) had matching payments numbers in both datasets, while 335 (78.8%) had a greater number in industry data (table 3, online supplementary file 5). In relative terms, 225 (52.9%) recipients had payments reported only in industry data (the relative difference of 100%), while 81 (19.1%) had a relative difference lower than 50%.

Of 425 recipients, 416 (97.9%) had payments with a value greater than 0 in at least one dataset (online supplementary file 5). Of those, 356 (85.6%) had a higher value of payments in industry data, and 8 (1.9%) had matching payment values. In relative terms, 295 (70.9%) recipients had payments reported exclusively in industry data, and 73 (17.5%) recipients had a relative difference lower than 50%.

Overall, recipients with a high overall number and value of payments usually had high absolute differences (online supplementary file 5). Relative differences were lower for recipients with higher overall number and value of payments.

The shares of recipients with matching numbers and values of payments showed no improvement over time, but the shares of those with relative differences lower than 50% increased (from 13.7% to 17.3% and from 9.7% to 14.5%, respectively) (online supplementary file 6).

## Missing payments

After analysing the pattern of absolute and relative differences we now focus on donors and recipients with the relative difference of 100%, that is, those with payments reported only in either industry or patient organisation data.

## Industry data

There were 24 donors, representing 38.1% of those in industry data, which seemed not to comply with the ABPI Code[13] as their payments were reported exclusively in

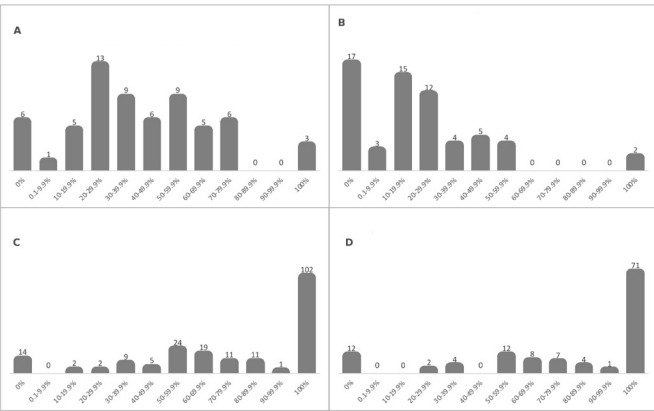

**Figure 2** (A) Overlap between recipient lists and patient organisation data—based on the number of payments. (B) Overlap between recipient lists and patient organisation data—based on the value of payments. (C) Overlap between donors lists and industry data—based on the number of payments. (D) Overlap between donors lists and industry data—based on the value of payments. The bars in (A) and (B) show the number of donors with varying level of overlap between their recipient lists and patient organisation data based on the number (A) and value (B) of payments. For example, based on the number of payments there were 13 companies with 20%–29.9% recipients also being reported in patient organisation data (A). The bars in (C) and (D) show the number of recipients with varying level of overlap between their donor lists and industry data based on the number (C) and value (D) of payments. For example, based on the number of payments there were 24 recipients with 50%–59.9% of donors also being reported in industry data.

patient organisation data over the entire period of observation (online supplementary files 7, 8, 9). The number and value of payments associated with these donors (in patient organisation data) were 128 and £1 610 321.1, equivalent to 3% of the number and value of payments in industry data. The annual shares of missing donors and their payments were considerably higher as some companies did not report payments only in some years. The annual shares of missing payments were decreasing over time.

### Patient organisation data

There were three donors, constituting 3.6% of those in patient organisation data, whose payments were only reported in industry data (online supplementary files 10, 11, 12). The number and value of payments associated with these donors (in industry data) were 9 and £92 208.2, constituting 0.5% and 0.3% of the number and value of payments in patient organisation data. As with industry data, the annual shares of missing donors and their payments were higher as some donors were not reported in patient organisation data only in some years. The annual shares of missing payments were decreasing during the study period.

There were 225 recipients missing from patient organisation data, representing 112.5% of recipients in this dataset (online supplementary files 13, 14, 15). The number and value of missing payments were 1472 and £14 023 475.41 (88.6% and 42.4% of the respective figures in patient organisation data). The yearly share of missing recipients and their payments increased, while the yearly share of missing payment values decreased.

### Agreement between donor–recipient ties reported by industry and patient organisations

After describing the under-reporting as a donor/recipient characteristic we now consider it by examining the distribution of ties between donors and recipients in each dataset

There were 1101 ties between donors and recipients formed by at least one payment in industry data and 626 in patient organisation data (online supplementary file 16). In 428 instances both datasets agreed about the donors and recipients, representing 38.9% and 68.4% of ties in each dataset, respectively. Of these, in 87 (20.3%) instances both donors and recipients reported the same number of payments. The number of payments constituting those matching ties was 162, equivalent to 3.8% and 9.8% of the total in industry and patient organisation data, respectively.

There were 1088 ties involving payments with a value greater than 0 in industry data and 326 in patient organisation data. In 34 instances, both datasets agreed about donors and recipients (3.1% and 10.4% of ties in each dataset). Of these, in 20 instances (58.8%) both sides provided matching payment values. The ties with matching values were worth £325 108.4, representing 0.6% and 1.0% of the value of payments in industry and patient organisation data, respectively.

The proportion of shared ties remained stable across time (online supplementary file 16).

### Overlap between recipient and donor lists in the two datasets

Having compared the distribution of ties in each dataset, we now consider ties as a characteristic of donors and recipients. Specifically, we analyse under-reporting by comparing donor and recipient lists in each dataset against the other dataset.

#### Overlap between recipient lists and patient organisation data

Of 63 donors reporting at least one payment in the industry dataset, only 3 (4.8%) had all their recipients mentioning them as donors, the situation representing the full overlap between recipient lists and patient organisation data (figure 2A). Conversely, the recipient lists of 6 (9.5%) donors had 0 overlap with patient organisation data. The recipient lists of the remaining donors were situated between the two extremes, with 40 (63.5%) donors having less than 50% overlap, meaning that only half of the recipients were reflected in patient organisation data.

Considering the recipient lists calculated using the value of payments (62 donors reporting payment values greater than 0), the full and 0 overlap occurred for 2 (3.2%) and 17 (27.0%) donors, respectively (figure 2B).

Overall, 56 (90.3%) donors had less than 50% overlap with patient organisation data.

## Overlap between donors lists and industry data

Of the 200 recipients in patient organisation data, 102 (51.0%) had all donors mentioning them as recipients, which represented the full overlap between donor lists and industry data (figure 2C). Contrastingly, the donor lists of 14 (7.4%) recipients had no overlap with industry data. Overall, 32 (16.0%) recipients had less than 50% overlap with industry data.

Based on the value of payments (121 recipients reporting payment values greater than 0), the full and 0 overlap occurred for 71 (58.7%) and 12 (9.9%) recipients, respectively (figure 2D). There were 18 (14.9%) donors with less than 50% overlap with industry data.

Over time, there was little improvement in the share of patient organisations reporting funding from drug companies listing them as recipients (online supplementary files 17 and 18). On the other hand, donors were reporting an increasing share of patient organisations mentioning their funding (online supplementary files 19 and 20).

## DISCUSSION
### Principal findings

Our study confirms that earlier concerns about the under-reporting of payments in relation to specific drug companies[15] and charities[36 37] in the UK were not isolated instances. It demonstrates that neither the industry self-regulatory system for disclosure of payments to patient organisations[1] nor a state-run system built for broader financial transparency purposes prevented a high extent of under-reporting. Importantly, the under-reporting occurred despite the media and research salience of the transparency of financial relationships between the industry and patient organisations.[19 38–40]

Although some yearly variation might be expected given differences between when a payment was made and when it was received or spent, the overall large and seemingly increasing discrepancies between industry and patient organisation records are concerning. The number and value of payments disclosed by drug companies were likely to be higher because the ABPI Code has a formal definition of a payment and stipulates that payment values should be recorded whenever possible.[13] Charity regulators lack similar specific requirements.[41] Further, some patient organisations may redistribute money to other collaborators and therefore they may report smaller amounts than donating drug companies.[42] Nevertheless, any differences between donor and recipient records should be minimised by the requirement for a written contract between the parties, introduced by the ABPI Code, implying a shared understanding of how much is paid, to whom and how.[13]

What also indicates under-reporting is that few companies and patient organisations had matching—or even broadly similar—records in the two datasets, with many having differences exceeding hundreds of payments or millions of pounds. Both drug companies and patient organisations had a larger number of payments in the other dataset, but for both sides the value of payments was greater in industry data. The contrast between the datasets was higher for patient organisations, with a majority having a greater number and value of payments reported in industry data. This pattern suggests inadequacies of the existing charity regulator governance of the reporting of corporate payments. It further corresponds with the results of a recent journalistic investigation into the under-reporting of payments by a major charity,[36 37] prompting a rebuke from a charity regulator.[43]

We also unearthed unreported payments associated with companies missing from either dataset. The under-reporting of payments is further indicated by the fact that donor–recipient ties reported in both datasets were in minority. This was particularly the case for ties involving matching payment numbers or values. The extent of the overlap between donor and recipient lists reported in the two datasets was similarly limited. Therefore, donors and recipients rarely disclosed payments reported by the other side.

As drug companies from our sample signed the ABPI Code, missing payments indicate that they either did not meet the obligation to disclose or removed their online disclosure reports. We could not find any evidence of specific obligations for UK charities to name funders in their accounts.[41] Their absence is motivated by concerns that 'the loss of donor anonymity would result in a decrease in voluntary income reported by charities. There were also concerns about the practical implications and about how much interest "general users" of charity accounts would have in this disclosure'.[44] Nevertheless, any unreported payments could be misleading for patient organisation members, supporters, expert bodies relying on patient testimonies,[21] policymakers and the public.[36 43] Unreported payments may indicate a culture of corporate manipulation of patient organisations.[45 46] Indeed, an oft-used argument for increasing transparency of payments is that 'sunshine' mitigates against misbehaviour and undue influence.[47] Crucially, despite challenges in achieving financial sustainability,[48] some patient organisations do not accept industry funding.[49 50]

Finally, under-reporting should be minimised by codes and reporting standards seeking to ensure faithful disclosure. The ABPI's self-regulatory authority, the Prescription Medicines Code of Practice Authority (PMCPA), is tasked with ensuring compliance under the ABPI Code, including through guidance and training, occasional active monitoring, and with the possibility to sanction companies that breach their obligations.[51 52] Likewise, the UK Charity regulators monitor a selection of annual accounts annually and, in instances of inaccurate reporting, will contact the charity directly to provide advice or request the accounts be resubmitted.[53–55]

More broadly, a similarly high extent of under-reporting was found in Italy, with less than a third of patient organisations identified as funding recipients disclosing industry funding.[19] High levels of under-reporting have also been identified internationally in relation to other recipients of industry payments, including authors of clinical practice guidelines[23 24] and scientific publications.[26 27 29]

### Strengths and limitations

This is the first study examining the under-reporting of payments to patient organisations using large samples of donors and recipients over a period of time. It systematically compares the under-reporting by the two sides using five complementary measures. These measures account for the varying nature of different types of payment data and therefore can be replicated elsewhere.

Our study has several limitations. First, while Disclosure UK covers a vast majority of the UK pharmaceutical industry,[14] our sample was not exhaustive, as demonstrated by additional companies found in patient organisation accounts. Second, excluding companies outside of our sample is likely to have underestimated the under-reporting but reaching data saturation would require several further rounds of data scrapping. Third, we identified patient organisations using drug company disclosure reports, but sampling starting from patient organisations could have produced different results. Fourth, patient organisations registered with charity regulators but with yearly income lower than £25 000 are not required to submit annual accounts. Therefore, we might have excluded, entirely or in specific years, some of the patient organisations identified as recipients of industry funding, if their yearly incomes were below that threshold. Fifth, some patient organisations were excluded due to the conversion of financial years into calendar years, which might have increased discrepancies with drug company disclosure reports. Sixth, some companies might have reported payments from 2016 in subsequent years, but the extent of delayed reporting is likely to have been minimal,[1] and there was no delayed reporting in patient organisation annual accounts. Seventh, we only considered annual accounts submitted to charity regulators, while funders might have also (or instead) been disclosed on the charity's website. However, website disclosures are likely to undergo frequent changes, and often lack transparency.[16–20] Eight, evaluating the extent of under-reporting precisely is impossible because no definitive list of payments exists; consequently, there could be payments undisclosed by both sides. Finally, considering payment descriptions could reveal further discrepancies, including different payment goals reported by donors and recipients; it could also identify payments made via third parties (eg, public relations companies[43]) or benefiting patient organisations indirectly.[15]

## CONCLUSIONS AND POLICY IMPLICATIONS

Although the full scale of under-reporting of industry payments to patient organisations remains unknown, it concerns both donors and recipients, and involves a considerable number and value of payments. Our findings put a question mark over the key claim that—at least in their current form—publicly available payment disclosures effectively address concerns about COIs resulting from industry payments.

We provide evidence for developing easily achievable improvements in the reporting of payments by both donors and recipients. Consistent with the ABPI Code, the ABPI's self-regulatory authority, the PMCPA, should investigate any instances of missing payments among the companies that have ratified the Code (data never published or prematurely removed from the public domain), and, if appropriate, penalise companies concerned in accordance with its mandate and available financial (so-called administrative charges) and non-financial (mainly naming and shaming) sanctions.[51 52] Separately, drug companies should improve data presentation following earlier recommendations,[15] especially standardisation of reporting and elimination of payments with no assigned values. Companies should store data for at least as long as is required by the Charity Commission for England and Wales, that is 5 years. Reduction in under-reporting could also be achieved if the ABPI started to publish yearly reports summarising payments to patient organisations disclosed by companies, as the ABPI currently does with payments to healthcare professionals and organisations in the Disclosure UK database.[32 33] Indeed, at least one European country—Sweden—has an industry-run, centralised disclosure database of payments to patient organisations,[56] and there is no reason why the ABPI should not have the same.

Given the shortcomings of the industry self-regulatory system, charity regulators should introduce tailored solutions related to reporting corporate funding in annual accounts, including a standardised template comprising a short payment description (including payment form and goal), its value and donor name. Patient organisation websites should report this information separately or include clear signposting to the annual accounts. A key step in refining these solutions would involve in-depth exploration of perspectives of patient organisations. However, ultimately, only a single state-run permanent database integrating payments reported by drug companies and patient organisations could eliminate under-reporting.

**Contributors** PO conceived and designed the study, managed, analysed and interpreted the data, as well as drafted the article. MC analysed the data and drafted the article. ER collected and managed the data, and drafted the article. SM designed the study, contributed to interpretation, drafted the article.

**Funding** SM and PO's work on this study was supported by the grant 'What can be learnt from the new pharmaceutical industry payment disclosures?' awarded by the Swedish Research Council for Health, Working Life and Welfare (FORTE #2016-00875). ER's work was funded through the above grant.

**Competing interests** None declared.

**Patient consent for publication**  Not required.

**Ethics approval**  The ethical implications of this study article were approved via a peer ethics review process at the Department of Social and Policy Sciences, University of Bath in April 2016. This study did not require a full ethical approval as it relied on publicly available data aggregated at the organisational level.

**Provenance and peer review**  Not commissioned; externally peer reviewed.

**Data availability statement**  All data relevant to the study are included in the article or uploaded as supplementary information. The authors of this study agree to share data underpinning this study in the form of web supplements available from the University of Bath Research Data Archive. The raw data poses no risk to anonymity of individuals as it draws on publicly available reports concerning financial transfers between organisations. The reference for this dataset is: Ozieranski, P., Csanadi, M., Rickard, E., Mulinari, S., in press. Underreporting of drug industry payments to patient organisations in the UK (2012-2016). Bath: University of Bath Research Data Archive. https://doi.org/10.15125/BATH-00784

**ORCID iD**
Piotr Ozieranski http://orcid.org/0000-0002-2023-3288

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
