## [Reviewer comments · BMJ Open]

ARTICLE DETAILS

TITLE (PROVISIONAL)	An underreported relationship: a comparative study of pharmaceutical industry and patient organisation payment disclosures in the UK (2012-2016)
AUTHORS	Ozierski, Piotr; Csanádi, Marcell; Rickard, Emily; Mulinari, Shai

VERSION 1 – REVIEW

REVIEWER	Joel Lexchin York University Canada
REVIEW RETURNED	08-Mar-2020

GENERAL COMMENTS	This study looks at underreporting of donations by pharmaceutical companies and patient groups in the United Kingdom. It is very data rich and that presented a problem for me in trying to keep all of the information straight. To this end, I wonder if Tables 1-3 could either be supplemented by Venn diagrams or else replaced in the text with Venn diagrams and the tables moved to Supplementary files. I also have some more specific comments: 1. Page 6, line 11: Who runs Disclosure UK, who administers the organization, does it have any annual reports?2. Page 6, line 18: Was the conversion based on the Bank of England's inflation index?3. Page 7, lines 34-38: It would be helpful for the authors to give specific examples of their calculations.4. Page 8, line 10: Were all 84 members of Disclosure UK?5. Page 21, line 19: The authors should mention whether compliance with the ABPI Code is actively monitored and what the sanctions are for breaching the Code.
---

REVIEWER	Paola Mosconi Istituto di Ricerche Farmacologiche Mario Negri IRCCS
REVIEW RETURNED	16-Mar-2020

GENERAL COMMENTS	This article reports additional data regarding the relationship between pharmaceutical companies and patient organization. The article compares payments disclosed by drug companies with those reported in the website of patient organizations. The sample is collected in UK where there is an important, long history of charity organizations, and drug companies (not all) recognised common code to disclose payments, value and purpose.
---

	This is not really a new issue, BMJ recently published a systematic review on this topic. Also, the methodology used in this research is not new or innovative. Nevertheless, it is important to continue to collect and publish new data, on different setting, and with different perspectives. In general, both, companies and associations, declare to be able to appropriately manage possible conflict of interest. Patient associations declare also that pharmaceutical, or device, companies are the only available source of funding, sometime the only one allowing to carry out activities, make information, or provide assistance. Nevertheless, there are also good examples of patient associations that have decided, and publicly declared, not to receive funds from for-profit groups showing that another way is feasible. The article of Ozieranski et al well describe the methods and the results obtained. However, it could have been even more remarkable considering also the type of initiatives supported. The simple analysis of the provided and received funds limits the impact of the research presented. It is important to understand the type of agreements between the parties, the influence that pharma funds have on the activities or on actions supported by patient associations. Associations are increasingly involved to participate in institutional groups, guideline, HTA exercises, research protocol or agenda, and, of course, information. Being independent is essential and should be guaranteed to society and patient represented. Another point to be clarified is why associations are still so little transparent. Policy of transparency together with codes of conduct are certainly necessary. However, in the absence of public financing (or alternative support strategies) profit companies will continue to support patient associations. The problem is therefore to understand for which type of projects (information, assistance, promotion, etc.) funds are used, and the influence that donors have in relation to recipients. This is never ended story and the collection of new data can be useful in trying to discuss a solution, or a share possible compromise, to this situation. In particular, this problem needs to be repeatedly considered, or known, by society as the most immediate action to take is to foster to reach full transparency of funding - far to obtain as show also by the results of the article. The results reported in the article are therefore interesting, some points need to be better clarify: Abstract Results section: data from the two datasets need to be balance, in my opinion some more data regarding patient organizations are needed, reducing some data regarding pharma dataset. Methods It is not clear if all data were collected by two different assessors. As it not easy to collect this kind of data, or find it on the website, a double evaluation could be useful to guarantee a good quality of data collection. If no, explanation should be provided.
--	--

	Why authors did not involve representatives of society/lay people in their analysis or in the discussion of the results? This choice should be motivated and discussed also in the discussion section. Results This is a descriptive study and a lot of findings are reported in this section, and this something is confusing. I suggest to consider to present data in the table or in flow chart. Maybe a table more, or more data in the tables could be useful for readers. Minor. Table 1, I suggest to change the order of the different year evaluation, from 2016 to 2012 left to right.
--	--

VERSION 1 – AUTHOR RESPONSE

Responses to Reviewer #1

Thank you for reading our manuscript and taking the time to offer a number of very detailed and insightful comments. We present our responses to your points in the table below. Thank you for your help in improving our work.

	Reviewer's comments	Authors' responses (manuscript version with accepted changes)
1	This study looks at underreporting of donations by pharmaceutical companies and patient groups in the United Kingdom. It is very data rich and that presented a problem for me in trying to keep all of the information straight. To this end, I wonder if Tables 1-3 could either be supplemented by Venn diagrams or else replaced in the text with Venn diagrams and the tables moved to Supplementary files	Thank you for this important suggestion. To clarify the relationships between different types of information reported in the paper we have now summarised all our outcome measures and the corresponding sections in the Results in Figure 1 (see p. 6). We hope that this figure will make relationships between the different outcome measures clearer as well as aid the navigation through the Results section. We think it is important to keep Tables 1-3 in the main body of the article as we believe they include information which is crucial for understanding our argument. However, we have decided to turn Table 4 into a Web Supplement (Web supplement 16). This table was not necessary to convey the essential points made in the relevant part of the Results.

2	Page 6, line 11: Who runs Disclosure UK, who administers the organization, does it have any annual reports?	Thank you for making this very useful point. We have now clarified (p. 4) that Disclosure UK is managed by the ABPI. We have also referred to the issue of annual reports and penalties for underreported payments (pp. 18-19).
3	Page 6, line 18: Was the conversion based on the Bank of England's inflation index?	Thank you for helping us clarify this point. We used the Consumer Price Index (CPIH) published by the Office for National Statistics to adjust the payment values for inflation. We have now cite the data source in the methods section (p. 4) and provide the specific CPIH values used for adjustments as a note under Table 1 (p. 8)
4	Page 7, lines 34-38: It would be helpful for the authors to give specific examples of their calculations	Thank you for this comment. Interested readers can verify how we made our calculations for each donor and recipient by consulting data provided in web supplements 3-20, including detailed breakdowns of absolute and relative differences between the two datasets (web supplements 3-6). We now clarify our signposting by adding an additional sentence in the Methods section (p. 5).
5	Page 8, line 10: Were all 84 members of Disclosure UK?	Yes, all the 84 companies were members of Disclosure UK, which directly follows on from the exclusion of non-Disclosure UK members from analysis, which we mention in the Methods section (p. 4). In addition to the 84 Disclosure UK members in patient organisation data we also found 61 non-Disclosure UK participants. This is of course a big concern but looking into these companies would require a stand-alone paper.
6	Page 21, line 19: The authors should mention whether compliance with the ABPI Code is actively monitored and what the sanctions are for breaching the Code.	We have clarified this issue on p. 18.

Responses to Reviewer #2

Thank you for reviewing our work so carefully and offering a range of very useful comments. We present our detailed responses in the table below. We appreciate your help in enhancing the quality of our work.

	Reviewer's comments	Authors' responses (manuscript version with accepted changes)
1	This is not really a new issue, BMJ recently published a systematic review on this topic. Also, the methodology used in this research is not new or innovative. Nevertheless, it is important to continue to collect and publish new data, on different setting, and with different perspectives.	Thank you for this comment. We now cite the newly published BMJ systematic review,¹ which was not available at the time of submitting the original version of the manuscript. We also cite an important Italian study² which, to the best of our knowledge, was the first one to compare the content of disclosures made by the industry and patient organisations. We also wish to respond to your comment about our study's methodology not being "new or innovative". In short, our study is a step towards developing an innovative methodology for measuring the underreporting of industry payments based on data coming from different sources and in the absence of knowledge about of the exact number of payments (See also p. 18 in the manuscript). The innovative nature of our study is clear in the context of the recently published systematic review,¹ which only considered patient organization-reported funding. The systematic review did not (1) look at industry reported-funding or (2) compare industry reported and patient organisations-reported funding. Against this background, we can say that our study calls into questions some of the conclusions reported in the BMJ systematic review as it shows that patient organization-reported funding is a very problematic data source.

		Our study is also innovative in the context of the pioneering Italian research.² When comparing industry- and patient-organisation reported funding this study focused on the correspondence between donors and recipients. In contrast, our study deploys several measures of underreporting at three levels of analysis (dataset, organisational, payment). These measures are important in achieving triangulation of results as the actual number and value of payments as well as recipients remain unknown. Furthermore, in contrast to the Italian study we use data sources which can be deemed more reliable than information websites. Specifically, both industry disclosure reports and patient organisation financial reports are governed by explicit rules and regulations and there are sanctions for breaching them. In fact, drug company disclosure reports only became available in 2012, that is after the data for the Italian study was collected.
2	In general, both, companies and associations, declare to be able to appropriately manage possible conflict of interest. Patient associations declare also that pharmaceutical, or device, companies are the only available source of funding, sometime the only one allowing to carry out activities, make information, or provide assistance. Nevertheless, there are also good examples of patient associations that have decided, and publicly declared, not to receive funds from for-profit groups showing that another way is feasible	Thank you for this important comment. We now stress that there are examples of patient organisations which decided not to receive funding from for-profit entities (p. 18)
3	it could have been even more remarkable considering also the type of initiatives supported. The simple analysis of the provided and received funds limits the impact of the research presented. It is important to understand the type of agreements between the parties, the influence that pharma funds have on the activities or on actions supported by patient associations. Associations are increasingly involved to participate in institutional groups, guideline, HTA exercises, research protocol or agenda, and, of course,	We entirely agree with your point about the need to study the types of projects funded by drug companies. In fact, we have coded the goals of payments reported by drug companies and patient organisations separately using a coding framework we used elsewhere.³ We sought to include this information but eventually decided not to do because this type of analysis would require comparing

	information. Being independent is essential and should be guaranteed to society and patient represented.	several payment goals reported by both sides. This would have been impossible in the current table given the constraints around word count and table numbers. However, we intend to take up this topic in a follow-up paper. We note this issue as a limitation of the current paper, too (p. 19).
4	Another point to be clarified is why associations are still so little transparent.	Thank you for this question. We are equally puzzled by the apparent insufficient progress in increasing the transparency of funding reported by both drug companies and patient organisations. In the concluding section of this paper we formulate a number of policy recommendations which – if followed – might improve the current state of affairs.
5	Policy of transparency together with codes of conduct are certainly necessary. However, in the absence of public financing (or alternative support strategies) profit companies will continue to support patient associations. The problem is therefore to understand for which type of projects (information, assistance, promotion, etc.) funds are used, and the influence that donors have in relation to recipients	We allude to these issues when replying to your third comment above. In a separate study, we note possible risks associated with the bulk of industry funding supporting patient group activities such as lobbying or public relations, in the absence of alternative sources of public funding.³ In the current paper, we call for the establishment of a public database that could comprise payments reported by patient organisations and drug companies (p. 19). This database would be a key step in improving our understanding of the types of projects being funded.
6	This is never ended story and the collection of new data can be useful in trying to discuss a solution, or a share possible compromise, to this situation. In particular, this problem needs to be repeatedly considered, or known, by society as the most immediate action to take is to foster to reach full transparency of funding - far to obtain as show also by the results of the article.	We entirely agree with this comment, as demonstrated by the concluding section of our article (p. 19).

7	Abstract Results section: data from the two datasets need to be balance, in my opinion some more data regarding patient organizations are needed, reducing some data regarding pharma dataset	We believe the abstract refers to data from both datasets in equal measure. When describing participants we refer to drug companies (and their data) as donors and patient organisations (and their data) as recipients.
8	Methods It is not clear if all data were collected by two different assessors. As it not easy to collect this kind of data, or find it on the website, a double evaluation could be useful to guarantee a good quality of data collection. If no, explanation should be provided	We now clarify this issue towards the end of the Methods section (p. 4-5).
9	Methods Why authors did not involve representatives of society/lay people in their analysis or in the discussion of the results? This choice should be motivated and discussed also in the discussion section.	We explain this in the methods section (p. 6), and subsequently make a recommendation for increased utilisation of patient organisations' perspectives in the Concluding section (p. 19).
10	Results This is a descriptive study and a lot of findings are reported in this section, and this something is confusing. I suggest to consider to present data in the table or in flow chart. Maybe a table more, or more data in the tables could be useful for readers.	Thank you for your helpful suggestion. We now include a Venn Diagram (Figure 1) in the Methods section to explain relationships between different outcome measures (p. 6)
11	Minor. Table 1, I suggest to change the order of the different year evaluation, from 2016 to 2012 left to right	Thank you. We see how this suggestion could be helpful but we thought it would be best to keep the data presentation in line with other cohort studies published in BMJ Open.⁴

1. Fabbri A, Parker L, Colombo C, et al. Industry funding of patient and health consumer organisations: systematic review with meta-analysis. *BMJ* 2020;368:l6925. doi: 10.1136/bmj.l6925
2. Colombo C, Mosconi P, Villani W, et al. Patient organizations' funding from pharmaceutical companies: is disclosure clear, complete and accessible to the public? An Italian survey. *PLoS One* 2012;7(5):e34974.

3. Ozieranski P, Rickard E, Mulinari, Shai. Exposing drug industry funding of UK patient organisations. *BMJ* 2019;365:l1806. doi: 10.1136/bmj.l1806
4. Mulinari S, Ozieranski P. Disclosure of payments by pharmaceutical companies to healthcare professionals in the UK: analysis of the Association of the British Pharmaceutical Industry's Disclosure UK database, 2015 and 2016 cohorts. *BMJ open* 2018;8(10):e023094.

VERSION 2 – REVIEW

REVIEWER	Paola Mosconi Istituto di Ricerche Farmacologiche Mario Negri IRCCS
REVIEW RETURNED	01-May-2020
GENERAL COMMENTS	The reviewer completed the checklist but made no further comments.